# Cognitive Behavioral Group Therapy for Chronic Tinnitus in a German Tertiary Clinical Real-World Setting

**DOI:** 10.3390/ijerph20064982

**Published:** 2023-03-11

**Authors:** Martin Schecklmann, Franziska C. Weber, Astrid Lehner, Berthold Langguth, Stefan Schoisswohl

**Affiliations:** Department of Psychiatry and Psychotherapy, University Regensburg, Universitätsstraße 84, 93053 Regensburg, Germany

**Keywords:** tinnitus, cognitive behavioral therapy, loudness, annoyance, CBT

## Abstract

Cognitive behavioral therapy (CBT) was shown to be effective in reducing tinnitus-related distress in numerous controlled trials. Real-world data from tinnitus treatment centers are an important addition to controlled trials for demonstrating the ecological validity of the results from the randomized controlled trials. Thus, we provided the real-world data of 52 patients participating in CBT group therapies during the time period from 2010 to 2019. The groups consisted of five to eight patients with typical CBT content such as counseling, relaxation, cognitive restructuring, attention training, etc. applied through 10–12 weekly sessions. The mini tinnitus questionnaire, different tinnitus numeric rating scales and the clinical global impression were assessed in a standardized way and were analyzed retrospectively. All outcome variables showed clinically relevant changes from before to after the group therapy, which were still evident in the follow-up visit after three months. Amelioration of distress was correlated to all numeric rating scales, including tinnitus loudness but not annoyance. The observed positive effects were in a similar range as effects of controlled and uncontrolled studies. Somewhat unexpected was the observed reduction in loudness, which was associated with distress and the missing association of changes in distress with annoyance as it is generally assumed that standard CBT concepts reduce annoyance and distress, but not tinnitus loudness. Apart from confirming the therapeutic effectiveness of CBT in real-world settings, our results highlight the need for a clear definition/operationalization of outcome measures when investigating psychological interventions of tinnitus.

## 1. Introduction

Patients with chronic diseases (e.g., tinnitus) are seeking a cure. Even though numerous often-called promising treatment approaches have been published, none one of them offers a cure for subjective chronic tinnitus so far [1]. The stated reasons are the tinnitus heterogeneity and the complex pathophysiology involving the ear, auditory and non-auditory brain networks and the somatosensory system of the head and the neck [1]. Typically, abnormal auditory and somatosensory inputs trigger the dysfunctional neuronal compensation mechanisms in the auditory and non-auditory brain networks, which in turn lead to the phantom sound, chronification and burden [2,3]. Recent efforts from tinnitus experts led to the suggestion of tinnitus disorder as an independent diagnostic entity and should be differentiated from tinnitus itself [4].

A tinnitus disorder comprises psychological suffering with all the typical concomitants such as negative emotions, physiological hyper-arousal and dysfunctional thoughts [4]. The best message in the situation of a missing cure is that counseling and psychotherapy can efficiently help in coping with distressing tinnitus (Fuller et al. 2020). In standard care, psychotherapy for tinnitus is applied in weekly sessions over several weeks. Newer approaches are implementing these units through mobile applications. The best evaluated and most efficient treatment for chronic tinnitus is cognitive behavioral therapy (CBT) [5]. The efficacy was shown at the highest evidence level as indicated by several reviews and meta-analyses, including randomized controlled trials [5]. Interestingly, the studies conducted in a controlled design are very often published in contrast to non-controlled studies. This can be seen in 10 non-randomized studies that were not included in a recent Cochrane review in contrast to 28 included randomized trials. While inspecting the published manuscripts (80 articles with the search terms “cbt” and “tinnitus” in PubMed in July 2022; pubmed.ncbi.nlm.nih.gov), only three studies presented real-world data in a standard care out-patient approach of weekly group therapy sessions. These three articles with sample sizes of 68, 68 and 34, respectively, showed significant effects and were in line with the findings of a recently published article with data from over 300 patients. Real-world data (which we define as data not from controlled studies) are important as the data from randomized controlled trials are limited to the patients with characteristics that fulfil the usually strict inclusion criteria. Moreover, the patients are frequently specifically recruited for participation in clinical trials and, therefore, may differ in their tinnitus characteristics from the patients who are actively present in a tinnitus treatment center seeking help. Thus, complementing the data from randomized controlled trials (RCT) with real-world data is essential to ensure an ecological validity. Therefore, we present the real-world data from our out-patient CBT group setting in a tertiary clinic.

## 2. Methods

Group therapy consisted of 10–12 group sessions comprising five to eight patients, with one group session lasting 90–120 min. The patients were first seen in a general consultation visit at the Interdisciplinary Tinnitus Center of Regensburg (Germany). This visit consisted of a consultation in the ear–nose–throat department, including audiology plus a consultation in the psychiatry department. The consultation was based on the initiative of the patients or other physicians and meant that the patients were from outside the clinic. Patients with a recommendation for CBT as indicated by an interdisciplinary team meeting were again interviewed 1–4 weeks before the group therapy started and were also screened for group therapy eligibility in a clinical context via interviews. The criteria were that the patients needed to be at least moderately suffering based on the clinical impression and show group compatibility, manageability of the time schedule of the group visits, no relevant psychiatric comorbidity and assured compliance. Five of these patients had a diagnosis of depression (diagnosed from a clinical interview conducted by psychiatrists or psychotherapists in training), but these patients were stable and not severely ill, which guaranteed their ability to join the group sessions. In this intake visit or preliminary talk, the patients were informed about the aims and procedures of the therapy. The group therapy took place on a weekly basis (same weekday and daytime) over a course of three months. Three months after therapy end, a booster session (relaxation, repetition, outlook) took place.

The group concept was based on three different German manuals of CBT for chronic tinnitus [6,7,8]. The contents of the groups were counseling with a focus on tinnitus and hyperacusis, a definition of therapeutic aims, relaxation (each session started with different relaxations exercises: breath and muscle relaxation, imagination), mindfulness-based input, a disease model based on a vicious circle, techniques of cognitive restructuring (ABC (activating event → beliefs → consequences) model), reframing, automatic thoughts, tinnitus myths, thought stopping, functional symbols (comfortable sound generators) for tinnitus, activation strategies, attention/distraction techniques, stress and sleep management and an exposition exercise (forced perception of tinnitus) at the end of the therapy. Contents of the group were minorly adjusted every year. The groups were moderated by one or two therapists who were psychotherapists in training. The group sessions were conducted under supervision.

Patients filled in questionnaires during four clinical visits. The first visit took place several weeks to months before the group therapy started (general consultation; screening). The second visit was equivalent to the intake (preliminary talk, baseline) or the first group therapy session. The third visit was at the last group therapy session (three months later, treatment end). The follow-up visit was a booster group session three months after the group therapy. The questionnaires used during all the study visits were the mini tinnitus questionnaire (miniTQ) [9], consisting of 12 items that measured tinnitus distress and provided numeric rating scales for tinnitus loudness, discomfort, ignorability, annoyance and unpleasantness on a scale from 0–10 [10]. The clinical global impression (CGI) had to be filled out at the treatment end and during the follow-up visit. Data with respect to hearing loss (mean average hearing loss left ear: 20.81 ± 16.49 dB HL; right ear: 18.53 ± 13.07; based on a standard audiogram with frequencies tested from 125 Hz to 8 kHz in octave and semi-octave steps), tinnitus duration (171.70 ± 86.25 months), age (59.52 ± 11.28 years) and gender (17 females) were extracted from the first consultation visit.

All data were collected and analyzed within the framework of the Tinnitus Research Initiative database [10], which was approved by the ethics committee of the University of Regensburg, Germany (ethical approval number: 08/046). In this retrospective analysis data of 52 patients were included. Overall, 73 patients in 13 group therapies partook in the group sessions, but 21 did not fill in the questionnaires completely, and thus were excluded from the present analysis. The therapeutic success was analyzed descriptively by showing the changes in the tinnitus grades based on the miniTQ from the baseline to the treatment end visit using a Sankey plot with the following categories: compensated (1–7 points), moderately affected (8–12 points), severely affected (13–18 points) and extremely affected (19–24 points). Linear mixed effects models were deployed for a statistical evaluation of the therapeutic success over the four study visits (screening, baseline, treatment end, follow-up) per questionnaire. The time was handled using a fixed approach and the individual patient was given a random effect. The fixed effect of time was evaluated by the expected mean square approach, and in case of an effect, post-hoc Tukey contrasts were used to follow up on potential score differences between study visits. In addition, Spearman correlations were conducted between tinnitus distress (miniTQ) and the five numeric rating scales to evaluate whether tinnitus distress changes (post—pre score) were in accordance with other subjective markers of tinnitus burden. Correlation results were corrected for multiple testing using the Bonferroni method. All statistical analyses were executed with the software R (R version 4.0.3; R Foundation for Statistical Computing, Vienna, Austria). Significance levels were set at the 5% level for all analyses.

## 3. Results

Based on the Sankey plot, as shown in Figure 1, the number of patients who were extremely and severely affected was reduced from baseline to treatment end. Accordingly, the number of patients who were moderately affected or compensated increased. Based on the CGI at treatment end, all patients except two (*n* = 1: no change; *n* = 1: very much worse) showed an amelioration of their symptoms (*n* = 13: very much better; *n* = 19 much better; *n* = 16: minimally better). For two patients, we had no information. The effects were comparable during the follow-up visit (*n* = 13: very much better; *n* = 14 much better; *n* = 18: minimally better; *n* = 2: no change; *n* = 1: very much worse; *n* = 4: no information).

For the linear mixed effects analysis, we found significant fixed effects of time for all six dependent variables (Figure 2). Post-hoc tests showed the same pattern of change over time for all dependent variables (Figure 2): no change from screening to baseline, a significant decrease from baseline to end of treatment and follow-up and no difference between end of treatment and follow-up.

All the correlations between tinnitus distress (miniTQ) and numeric rating scales were significant (also at corrected level) except for the correlation with tinnitus annoyance (Figure 3).

## 4. Discussion

Group CBT for tinnitus in a real-world setting within a tertiary psychiatric clinic was shown to be effective based on the amelioration of all the symptoms in the majority of the patients as measured by the miniTQ (*p* < 0.001), numeric ratings scales (*p* < 0.001) and CGI (improvement in 92% of the patients). CBT is the best evaluated therapeutic concept for patients suffering from chronic tinnitus with the highest evidence in the field of tinnitus [11]. In comparison to other interventions—such as non-invasive brain stimulation which has been performed regularly in Regensburg—the effects were much higher [12]. Additionally, a retrospective analysis over a five year period (patients visiting a tinnitus consultation in the last five years surveyed at one specific time point) showed that the tinnitus distress as measured with the tinnitus questionnaires was improving over time [13]. Again, the changes were far below the effects of the present group therapy. The Cochrane-based meta-analysis showed an amelioration in the THI [5] scores of approx. 11 for CBT in contrast to a waiting list or no intervention and of about six THI points in contrast to active control conditions, including interventions such as relaxation, information and internet-based discussion forums. The THI was highly correlated to the tinnitus questionnaire [14] and the miniTQ was an extract of 12 items out of the tinnitus questionnaire [9]. The THI had a four-fold greater range from 0–100 in contrast to the miniTQ which had a range of 0–24. The change from pre to post-intervention in the present study was 5.98 points in the miniTQ, which showed that real-world CBT was at least as effective as the controlled conditions of clinical trials. Our real-world data were also in line with four other papers reporting real-world data, which all demonstrated a significant reduction in the tinnitus burden [15,16,17,18].

According to the mini tinnitus questionnaire data, the clinical global impression was also dramatically increased after therapy. Approx. 92% of the patients showed a treatment response at the end of the therapy and 86% after three months. The CGI was conducted as patient rating for the global improvement and represented not only the tinnitus distress but also the global wellbeing, thus highlighting the general value of CBT for tinnitus in functional disability. The minimal clinically important difference for the THI was an amelioration of six or more points and a normalization to the CGI [19]. This again highlighted the relevance of CBT in tinnitus for clinical significance and not only for statistical significance.

Apart from the miniTQ score, most of the other subjective parameters of tinnitus measured using the numeric analogue scales also decreased and were highly correlated with the miniTQ change. This means that the changes assessed with the different scales were comparable within the single patients. The higher the miniTQ reduction at the end of treatment, the better all the other measures. Most of these parameters were distress-related, such as discomfort, ignorability and unpleasantness, and it is highly considerable that these assessments were related. However, there was also a reduction in the score of the numeric rating scale for the loudness, and this reduction was highly correlated with the other outcome measures. This finding was somewhat surprising since the concept of CBT for tinnitus was not focused on a decrease in the tinnitus loudness but on an improvement in coping with tinnitus. It is often very difficult for patients to clearly differentiate between distress and loudness, so long as the loudness is measured on a self-rating level. In this context, the measurements of rather objective measures (e.g., audiological measures of minimal masking level or tinnitus matching) should be included in such trials.

The significant amelioration of the annoyance was not correlated with the amelioration of the tinnitus distress. This means that the group got better on all the rating scales, but the change of distress and annoyance was not in the same range for each patient. Annoyance is different from distress, unpleasantness, discomfort, ignorability and loudness as it is highly related to the emotion of anger. Several analyses and concepts related the distress of tinnitus with depression (sadness, helplessness etc.) and anxiety but ignored the additional basic emotion of anger. From our own clinical experience there are many patients who do not exhibit depression or anxiety. However, they complain that “it annoys/bugs/bothers me, or it sucks”. Some years ago, 719 international health care users with tinnitus, health care professionals, clinical researchers, commercial representatives and funders voted in a Delphi survey for “intrusiveness”, “acceptance of tinnitus”, “mood”, “negative thoughts and beliefs”, and “sense of control” as the outcome measures for the psychological interventions in tinnitus [20]. The next step after this survey was claimed to be the operationalization of these measures, which might focus on different emotional domains.

## 5. Conclusions

In sum, our real-world data highlighted the effectiveness of group CBT for tinnitus on a similar level as the controlled trials.

## Figures and Tables

**Figure 1 ijerph-20-04982-f001:**
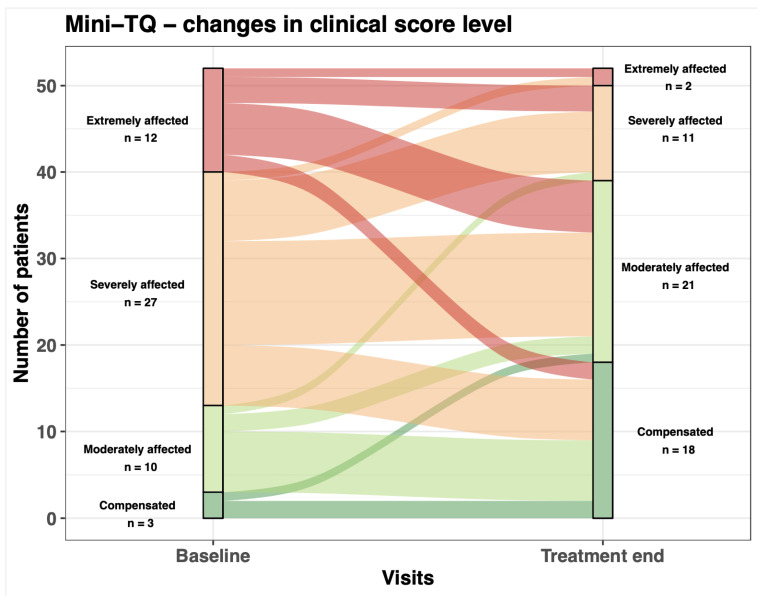
Changes in the tinnitus grades from before to after the group therapy. Sankey plot of changes in clinical score level based on the mini tinnitus questionnaire.

**Figure 2 ijerph-20-04982-f002:**
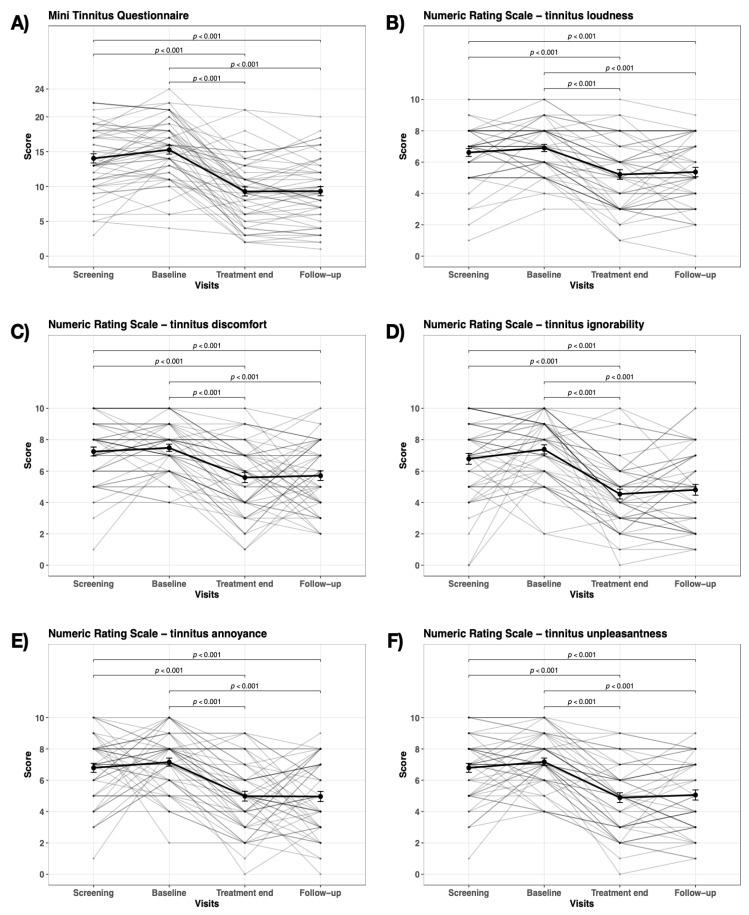
Changes in the tinnitus ratings. Changes in the tinnitus ratings over the four visits with cognitive behavioral therapy taking place between the baseline and the treatment end for the different measures of the tinnitus burden.

**Figure 3 ijerph-20-04982-f003:**
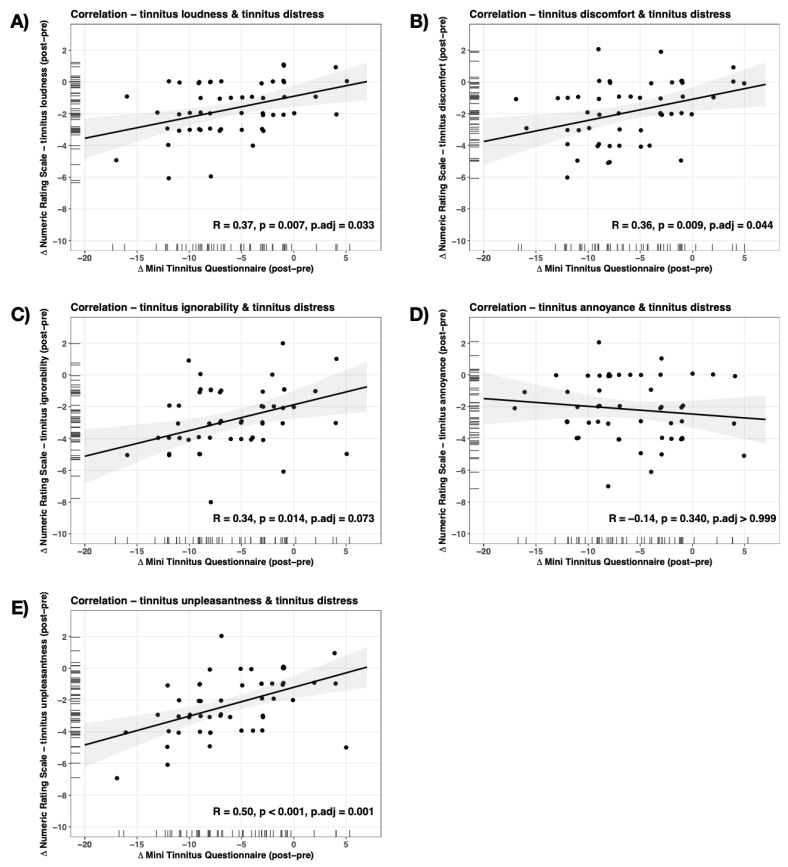
Correlation of changes from baseline to end of treatment of tinnitus distress with numeric ratings scales.

## Data Availability

This study does not report any data. The data can be made available to interested parties on demand.

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
