# Peer review of "Cognitive Behavioral Group Therapy for Chronic Tinnitus in a German Tertiary Clinical Real-World Setting"

_ijerph, 2023, doi:10.3390/ijerph20064982_

Round 1
Reviewer 1 Report
In the present retrospective study authors present real-world data from their out-patient Cognitive Behavioral Therapy group setting in a tertiary clinic. References are up-to-date and the language seems correct.
Real-world data are important as data from randomized controlled trials are limited to patients with charasteristics fulfilling the usually strict inclusion criteria. Patients are frequently specifically recruited for participation in clinical trials and therefore may differ in their tinnitus characteristics from patients who actively present in a tinnitus treatment center seeking help. Complementing data from randomized controlled trials with real-world data is essential to ensure ecological validity.
Summarizing, i'd like to see a perspective study taking into account of several clinical disease entities such as hysterical vertigo and peripheral neuropathy in diabetics in order to better describe and improve the design of the present clinical model.
Author Response
Dear reviewer,
thank you for your positive response. For sure it is desirable and valuable to include other entities such as hysterical vertigo and peripheral neuropathy in diabetics in tinnitus studies. We will incorporate these thoughts in our clinical assessement in the future.
Martin Schecklmann,
on behalf of all authors
Reviewer 2 Report
Dear authors!
I congratulate you on your work.
It is about cognitive behavioural therapy in connection with tinnitus. Real world data was collected retrospectively, which proved the effectiveness as well as randomised studies. Based on the evaluation of the data of own patients, it could finally be proven that cognitive behavioural therapy also leads to good results in the clinical setting. Although it does not close any gaps through the use and evaluation of own data, it complements the work in this field. From my point of view there are no suggestions for improvement on methodology. The conclusions are in line with the formulated research question. The literature citations used correspond to the current state of the art.All illustrations are clearly and unambiguously labelled.
Author Response
Dear reviewer,
thank you for your positive response.
Martin Schecklmann,
on behalf of all authors
Reviewer 3 Report
This article reports on results of CBT as treatment for "moderate suffering from" tinnitus. It was well organized and well-written.
Methods:
line 76. What is meant by "no relevant psychiatric morbidity"? Since it is well-known that there is a high association of tinnitus with depression and anxiety, how were these patients eliminated? And if so, did that skew the results? Also, how were they assessed, i.e., which instruments were used to rule these out?
There appear to be no controls, i.e., those who were affected and were not treated with CBT - authors should address happen to these tinnitus sufferers over the same period of time with either no treatment or other types of treatment.
Results: Authors should quantify the results with effects of treatment, p-values, or 95% confidence intervals in their written explanation of results mirroring the Figure results, allowing the reader to focus on what the authors consider the most important findings.
Discussion:
Since these were patients from a psychiatric clinic, the question becomes - what psychiatric diagnoses did these patients have? Did these patients attend psychiatric counseling as well as CBT - i.e., all of them, some of them?
Do the authors recommend CBT just for those without psychiatric diagnoses? Once again, how would controls fair who had the same or no psychiatric diagnoses without the added treatment of CBT?
Author Response
Dear reviewer,
thank you for your positive and helpful comments. See below our item-to-item responses to your suggestions.
Martin Schecklmann
on behalf of all authors
Comment: Methods: line 76. What is meant by "no relevant psychiatric morbidity"? Since it is well-known that there is a high association of tinnitus with depression and anxiety, how were these patients eliminated? And if so, did that skew the results? Also, how were they assessed, i.e., which instruments were used to rule these out?
Response: We had five patients with depression in this sample. Evaluation of psychiatric status is done by psychiatrists or psychotherapists by training. This is done by clinical interview. As this is only a minor group and we assured that they can participate in the group therapy because the depression is not the main issue we abstained from analysing this sub-group. We include this information now in the manuscript: "Five of these patients had a diagnosis of depression (diagnosed by clinical interview of psychiatrists or psychotherapists by training), but these patients were stable and not severe ill which guaranteed ability to join the group sessions."
Comment: There appear to be no controls, i.e., those who were affected and were not treated with CBT - authors should address happen to these tinnitus sufferers over the same period of time with either no treatment or other types of treatment.
Response: Indeed, we have no controls as the analysis was retrospective. Also looking in data of other patients we have not the same time schedule. This we can not do direct comparisons with a control group. We included the following statement in the discussion: "In comparison to other interventions - such as non-invasive brain stimulation which has been done regularly in Regensburg - effects are much higher [12]. Also a retrospective analysis over a five year period show that tinnitus distress is getting better over time [13]. Again the changes are much below the effects of the present group therapy."
Comment: Results: Authors should quantify the results with effects of treatment, p-values, or 95% confidence intervals in their written explanation of results mirroring the Figure results, allowing the reader to focus on what the authors consider the most important findings.
Response: We included statistical data in the first sentence of the discussion which summarizes the findings: "Group CBT for tinnitus in a real-world setting in a tertiary psychiatric clinic was shown to be effective - based on amelioration of all symptoms in the majority of patients as measured with miniTQ (p < .001), numeric ratings scales (p < .001) and CGI (improvement in % of the patients)."
Comment: Discussion: Since these were patients from a psychiatric clinic, the question becomes - what psychiatric diagnoses did these patients have? Did these patients attend psychiatric counseling as well as CBT - i.e., all of them, some of them?
Response: As stated above five of them had depression but were at stable treatment before and during the CBT. Patients participated only if the mental status was table which means that therapy for depression was stable during the CBT. A corresponding statement can already be found in the methods: "(at least moderate suffering based on clinical impression, group compatibility, time schedule of the group visits manageable, no relevant psychiatric comorbidity, assured compliance)".
Comment: Do the authors recommend CBT just for those without psychiatric diagnoses? Once again, how would controls fair who had the same or no psychiatric diagnoses without the added treatment of CBT?
Response: In the context of the study co-morbidities were allowed under the named circumstances. We think that psychiatric co-morbidities may be allowed but than group content and setting should be changed. But we can not conclude this from our data. Thus, we abstain from advices with respect to this issue in the manuscript.
Round 2
Reviewer 1 Report
Published in revised form
Author Response
Thank you for your positive response.
Reviewer 3 Report
Introduction - does not really describe why this study was done. I understand that "real world" patients should be examined for tinnitus, but the difference between this trial and others that apply CBT is not clear. I'm not quite sure what "real world" patients are. What are they in contrast to? Are these participants who were recruited from outside the clinic? Or were they referred from their primary care doctor? This is unclear.
Methods - line 75-78 and 104-107 are results, should be in results section. Lines 119-120 - not a sentence.
Results - Haven't really described in methods what the criteria are for extremely and severely affected.
Discussion - lines 165-166 - unclear sentence. The newly added information is highly confusing.
Line 171-2 should be in methods.
173-174 - "about 5-6 points" is not a statistical analysis.
Author Response
Dear reviewer,
thank you for your positive and helpful comments. See below our item-to-item responses to your suggestions.
Martin Schecklmann
on behalf of all authors
Comment: Introduction - does not really describe why this study was done. I understand that "real world" patients should be examined for tinnitus, but the difference between this trial and others that apply CBT is not clear. I'm not quite sure what "real world" patients are. What are they in contrast to? Are these participants who were recruited from outside the clinic? Or were they referred from their primary care doctor? This is unclear.
Answer: Real-world is related to the tertiary clinical real-world setting which means that patients were not selected by any strict inclusion or exclusion criteria. We try to answer your questions by adding the following sentences (quotation marks) in the introduction and methods:
Real-world data "(which we define as data not from controlled studies - which have strict inclusion and exclusion criteria)" are important as data from randomized controlled trials are limited to patients with characteristics fulfilling the usually strict inclusion criteria.
"Consultation is based on the initiative of the patients or of physicians and means that patients are from outside the clinic."
Comment: Methods - line 75-78 and 104-107 are results, should be in results section. Lines 119-120 - not a sentence.
Answer: Lines 75-78 and 104-107 are sample descriptions. In our opinion the result of the study is not which sample characteristics we have but the effects of the CBT. Thus, we do not change the position of these lines. Lines 119-120: We deleted the word "Thereby" and exchanged "treated" by "handled".
Comment: Results - Haven't really described in methods what the criteria are for extremely and severely affected.
Answer: We already defined this in methods: Therapeutic success was analyzed descriptively by showing changes of tinnitus grades based on the miniTQ from baseline to treatment end visit using a Sankey plot with the categories compensated (1-7 points), moderately affected (8-12 points), severely affected (13-18 points), and extremely affected (19-24 points).
Comment: Discussion - lines 165-166 - unclear sentence. The newly added information is highly confusing.
Answer: We added some information in the sentence: Also a retrospective analysis over a five year period "(patients visiting a tinnitus consultation in the last five years were surveyed at one specific time point)" show that tinnitus distress "as measured with tinnitus questionnaires" is getting better over time
Comment: Line 171-2 should be in methods.
Answer: This sentence helps to explain our findings. Thus, it is correct to state this in the discussion.
Comment: 173-174 - "about 5-6 points" is not a statistical analysis.
Answer: We inserted the exact value.